# The incidence of postoperative periprosthetic femoral fracture following total hip replacement: An analysis of UK National Joint Registry and Hospital Episodes statistics data

Jonathan Nicholas Lamb[1,2‡], Jonathan Thomas Evans[2,3,4‡]*, Samuel Relton[5], Michael Richard Whitehouse[2], J Mark Wilkinson[6‡], Hemant Pandit[7‡]

1 Centre for Hip Surgery, Wrightington Hospital, Wigan, United Kingdom, 2 Musculoskeletal Research Unit, Translational Health Sciences, Bristol Medical School, Southmead Hospital, Bristol, United Kingdom, 3 Exeter Medical School, University of Exeter, St Lukes Campus, Exeter, United Kingdom, 4 National Institute for Health and Care Research Exeter Biomedical Research Centre, Royal Devon University Healthcare NHS Foundation Trust and University of Exeter, Exeter, United Kingdom, 5 Leeds Institute of Health Sciences, School of Medicine, University of Leeds, Leeds, United Kingdom, 6 Division of Clinical Medicine, School of Medicine and Population Health, University of Sheffield, Sheffield, United Kingdom, 7 Leeds Institute of Rheumatic and Musculoskeletal Medicine (LIRMM), University of Leeds, C/O Chapel Allerton Hospital, Leeds, United Kingdom

‡ JNL and JTE share first authorship on this work. JMW and HP are joint senior authors on this work.
* j.t.evans@exeter.ac.uk

**Data Availability Statement:** Data cannot be shared publicly because they are protected as part of a mandatory national audit. Anonymised data are available from the NJR Research Sub Committee for researchers who meet the criteria

## Abstract

### Background

Postoperative periprosthetic femoral fracture (POPFF) after total hip replacement (THR) requires complex surgery and is associated with a high morbidity, mortality, and cost. Although the United Kingdom based National Joint Registry (NJR) captures over 95% of THRs treated with revision, before June 2023 it did not capture POPFF treated with fixation. We aimed to estimate the incidence and epidemiology of POPFF treated with either surgery in England.

### Methods and findings

We performed a retrospective analysis of a mandatory, prospective database (NJR) linked to Hospital Episode Statistics (HES). All linkable primary THRs between 01/01/2004 and 31/12/2020 were included. Revision or fixation of POPFF were identified using a combination of procedural and diagnosis codes.

We identified 809,832 THRs representing 5,542,332 prosthesis years at risk. A total of 5,100 POPFF were identified that had been surgically treated by revision, fixation, or both, and 2,831 of these fractures were treated with fixation alone, meaning 56% were not represented with revision data alone. The incidence of POPFF needing surgery was 0.92 (95% CI 0.90, 0.95) per 1,000 prostheses years. This incidence was higher in patients over the age of 70 at the time of primary THR (1.31 [95% CI 1.26, 1.35] per 1,000 prostheses years) and for patients who underwent THR for hip fracture (2.19 [95% CI 1.97, 2.42] per 1,000

for access to confidential data. The process for applying for the data underlying the results presented in the study are available from https://www.njrcentre.org.uk/research/research-requests/. HES Admitted Patient Care data was made available via same route under sublicensing agreement between NJR and NHS England.

**Funding:** This work was funded by Orthopaedic Research United Kingdom (ORUK) Grant number 540 - Epidemiology of periprosthetic femoral fracture. (JTE) MRW is Principal Investigator (PI) of the National Joint Registry lot 2 contract (statistical analysis) team which is hosted by his employer, the University of Bristol; JTE is a member of the lot 2 contract team. HP (as a Chief Investigator) receives institutional funding from Zimmer Biomet, Depuy Synthes, Allay Therapeutics, Paradigm Pharma and Invibio. In particular, Zimmer Biomet has funded University of Leeds in relation to the on-going and previous research in the field of peri-prosthetic fractures. This study was supported by the National Institute for Health and Care Research Exeter Biomedical Research Centre. This paper presents independent research supported by the National Institute for Health Research (NIHR) Leeds Biomedical Research Centre (BRC). This study was supported by the NIHR Biomedical Research Centre at the University Hospitals Bristol NHS Foundation Trust and the University of Bristol. The funders had no role in study design, data collection and analysis, decision to publish, or preparation of the manuscript.

**Competing interests:** HP is a NIHR senior investigator and MHRA advisor. In the latter role, he advised MHRA regarding the incidence and risk factors associated with the post-operative peri-prosthetic femoral fractures. In particular, this was to identify the risks associated with one of the commonly used stems (CPT manufactured by Zimmer Biomet). This was a voluntary role and no funding from MHRA was/is associated with it. HP (as a Chief Investigator) receives institutional funding from Zimmer Biomet, Depuy Synthes, Allay Therapeutics, Paradigm Pharma and Invibio. In particular, Zimmer Biomet has funded University of Leeds in relation to the on-going and previous research in the field of peri-prosthetic fractures. HP is a paid consultant to Zimmer Biomet for the work in the field of peri-prosthetic fractures and arthroplasty related teaching / training. HP has also received personal funding from the following industry organisations (unrelated to the topic of this work): Invibio, Medacta International, Smith and Nephew, MATOrtho, Microport, Teleflex, Allay Therapeutics and Paradigm Pharmaceuticals. HP

prostheses years). This incidence appears to be increasing year on year. The cumulative probability of sustaining a POPFF within 10 years of THR was 1% and over 15% of patients died within 1 year of surgery for a POPFF.

## Conclusions

To date, the incidence of POPFF may have been underestimated with over 50% of cases missed if the case identification in this study is correct. After including these cases, we observed that POPFF is the largest reason for major reoperation following THR and patients sustaining these injuries have a high risk of death. The prevention and treatment of POPFF and requires further resource allocation and research.

---

## Author summary

### Why was the study done?

- When the thigh bone supporting a hip replacement breaks because of injury, it is called a periprosthetic fracture of the femur. Patients are usually treated with major surgery, which is associated with significant risk of complications, high cost, and even death.

- Currently the best estimates of how often these injuries occur after hip replacement in England are made using data which only counts when fractures are treated with one surgical method, revision and therefore miss a large proportion of cases.

- This study is the first large study combining hospital data and the best currently used source to get a much more complete picture of when these fractures occur. In addition to revision, the study was able to capture the other common treatment method, i.e., fixation, thereby providing a comprehensive and more meaningful review.

### What did the researchers do and find?

- A very large data set of all hip replacements performed in England were matched to health data on hospital admissions to find patients who were treated with either exchange of implants (revision) and fixation of the fracture without exchange of implants.

- This study found that previous estimates of periprosthetic fractures only represented about half of the actual number treated in hospitals over the past 2 decades. When cases treated with revision and fixation are combined, periprosthetic fracture appears to be the most common cause of major re-operation after hip replacement.

- Patients who go on to sustain a fracture tended to be older and frailer at the time of the first hip replacement than those who do not go on to break. One in 25 patients with fractures died within 30 days of their operation, and the hospital length of stay was over 2 weeks.

has a stock option with Allay Therapeutics. HP holds two patents (through University of Leeds - in the field of implantable sensors and recharging apparatus). None of these roles / funding from any of the industry organisations (either to HP or to his institution) influenced the design; collection, analysis, and interpretation of data presented in this manuscript)

**Abbreviations:** HES, Hospital Episode Statistics; KM, Kaplan–Meie; NHFD, National Hip Fracture Database; JR, National Joint Registry; PEP-R, Patient Experience Partnership in Research; POPFF, postoperative periprosthetic femoral fracture; PTIR, prosthesis time incidence rate; THR, total hip replacement.

## What do these findings mean?

- These results suggest that periprosthetic fracture after hip replacement is the biggest problem facing patients with a hip replacement and the scale of the problem has previously been underestimated.

- Health care providers and researchers should prioritise the prevention and care for these vulnerable patients.

## Introduction

Total hip replacement (THR) is the mainstay of treatment for end stage osteoarthritis of the hip and, in general, is safe and effective [1]. While most hip replacements are expected to last over 25 years, failures do occur and this results in severe pain and disability for the patient as well as the need for revision surgery [2], which is costly to both the patient and the health service [3,4]. Hip replacements can fail and need revision for several reasons including infection, joint instability, wear and/or loosening of the hip replacement components, unexplained pain, or when the femur supporting the hip replacement fractures [5]. When this occurs, it is known as a postoperative periprosthetic femoral fracture (POPFF) and is sudden, painful, and debilitating [6]. In most cases, patients are unable to stand and will need to be admitted to hospital for urgent surgery to stabilise the hip replacement allowing the patient to stand and walk. This stabilisation can be in the form of revision to a different type of hip replacement, or retention of the existing implants and internal fixation. Qualitative work has told us that the impact of these fractures is sudden, resulting in patients feeling "powerless" and in "dreadful, dreadful pain" [6]. Patients are often left waiting for days for urgent surgery, not knowing what is going on while specialist equipment is made available and until appropriate surgeons are available [7]. Even after surgery, patients report a loss of independence and confidence in their own mobility which can go on for years [6].

The outcomes of all hip replacements in England and Wales have been continuously monitored as part of the mandatory National Joint Registry (NJR) since 2003 and at the time of the 2023 20th annual report, held information on 1,488,541 primary hip replacement procedures [5]. Until June 2023, the NJR only collected information on hip replacements that had been revised (with additional, removal or modification of THR components) and not those that had been internally fixed with the existing implant remaining in place. This means that research relying on this data was missing fractures treated by fixation alone. As a result, it is impossible to estimate the full scale of the problem which limits the ability to plan and adequately resource the departments managing these injuries [8]. In 2019, the National Hip Fracture Database (NHFD) started collecting information on POPFF and suggest that as many as 60% of POPFF that are treated with surgery are treated with internal fixation, suggesting that the number of patients sustaining these life changing injuries may be twice as many as previously thought [9]. Bottle and colleagues used disease coding to identify patients with any periprosthetic fracture and found large numbers of patients with periprosthetic fractures, and incidence increased over the study period. Over 85% underwent surgical procedure but only 1 in 10 underwent revision operations. Findings were limited by the fact that they could not identify which part of the body the fracture related to and that the international classification of diseases (ICD-10) code they used to identify fractures had only recently been introduced and therefore uptake

was unknown [10]. Researchers combining Swedish Arthroplasty Registry and routinely collected health data have estimated a rate of POPFF of 1.4 cases per thousand THR, but application of these estimates to a UK population is problematic where risk factors such as age, comorbidity, indication, and implant use are not equivalent [11,12].

The aim of this study was to identify reoperation for POPFF after primary total hip replacement using a linked implant registry and healthcare data set. This will allow us to investigate the incidence, changes in incidence, demographics of patients sustaining surgically managed POPFF, and outcomes after fracture including length of stay and mortality.

## Methods

### Ethics statement

This study was approved by the NJR research subcommittee, registration(s): Establishing risk profiles for incident periprosthetic fractures, associated health care costs, and projections for the future [RSC2017/20 HQIP REF: 198] and Periprosthetic fractures around primary total hip replacement [RSC2019/07].

We performed a retrospective review of prospectively collected data for all patients who had details of a primary THR submitted to the NJR. Data was accessed through the NJR research portal and analysed with R (4.2.0). Approval for the study and the planned methodology was granted by the NJR research subcommittee (RSC2017/20 and RSC2019/07) prior to data access and no data driven changes to analyses took place. Hospital Episode Statistics (HES) data is collected on NHS funded procedures performed in the NHS or independent sector in England but are not collected in Wales, Northern Ireland, or the Isle of Man. This study is reported as per the Reporting of studies Conducted using Observational Routinely collected health Data (RECORD) Statement (S1 Checklist).

The study population was all THRs implanted in the NJR from 1 January 2004 and 31 December 2020 with data linkable to HES and consent for their data to be used in research. Linkage was completed by the NJR and data were accessed through the NJR research portal. THRs with incomplete or inconsistent data or using metal-on-metal bearings (previously shown to demonstrate poorer survival) [13] were excluded, as were cases where the implant design was not known.

Patients were included if they had undergone primary THR between 1 January 2004 and 31 December 2020 and had undergone a first reoperation for POPFF after the day of primary surgery. A combination of ICD10 and OPCS4 codes in the HES data were used to identify fractures that had undergone surgery, occurred on the same side as the linked primary hip replacement and after the date of the primary (see S1 Table). This set of codes has not been independently validated for specificity in identifying POPFF. In addition, HES data were used to identify patients that had undergone revision THR and not been reported to the NJR by using the ICD10 and OPCS4 codes already provided by the NJR for use in the annual data quality audit.

### Statistical analysis

The primary outcome was incidence of surgically managed POPFF. This consisted of fractures around hip replacement femoral components that were treated with either revision of the implant, internal fixation, or a combination [13,14]. Incidence was estimated using a count of new cases per year or prosthesis time incidence rate (PTIR), which was number of new events per 1,000 years that a prosthesis has cumulatively been in place. The choice of primary outcome (either revision of any part of the THR for any reason (NJR definition) or internal fixation of a POPFF) was made in partnership with the Patient Experience Partnership in

Research (PEP-R) group. They advised us that the need to undergo any further surgery is the most important outcome to them regardless of the reason for the surgery [15]. Secondary outcomes included patient survival to an endpoint of POPFF, patient survival to an endpoint of mortality, length of stay, and inpatient bed stay cost. Survival until POPFF was calculated using the Kaplan–Meier (KM) method for all included THRs, censored either by death, or administratively on 31 December 2020 [14]. Mortality was estimated using the KM method for all patients in the study. Length of stay was estimated as the number of whole days between date and time of admission and time of discharge from hospital. Inpatient stay costs were estimated by multiplying total inpatient bed days by an estimated cost of £586.59 per bed day [16]. All analyses were unadjusted.

## Results

There were 1,128,684 primary THRs available for analysis of which 864,793 were linkable to the HES database by a unique national identifier. There were 38,703 hips excluded as they had a metal-on-metal bearing surface and 16,256 with an unknown implant design. A flow chart of data sources, inclusions, and exclusions can be seen in Fig 1.

A total of 5,100 POPFF were identified that had either been treated with revision surgery or internal fixation, and 2,831 fractures were treated with internal fixation meaning that 56% were not represented in NJR reports and analyses that captured revision only and in previous research based upon these data. There were 520 additional revisions identified using OPCS4 codes that were not already in the NJR, this is roughly in keeping with results of the NJR annual data quality audits [5].

Fig 2 shows the most common reasons for revision, classified into the subgroups used in the NJR annual report but calculated using our study data set. The grey bars in the figure represent revision operations and the black area represents the additional POPFF treated with internal fixation identified by this study.

### Demographics

The demographics of our study population (primary THRs that were linkable with HES data) were representative of the overall NJR population described by the NJR 2023 annual report

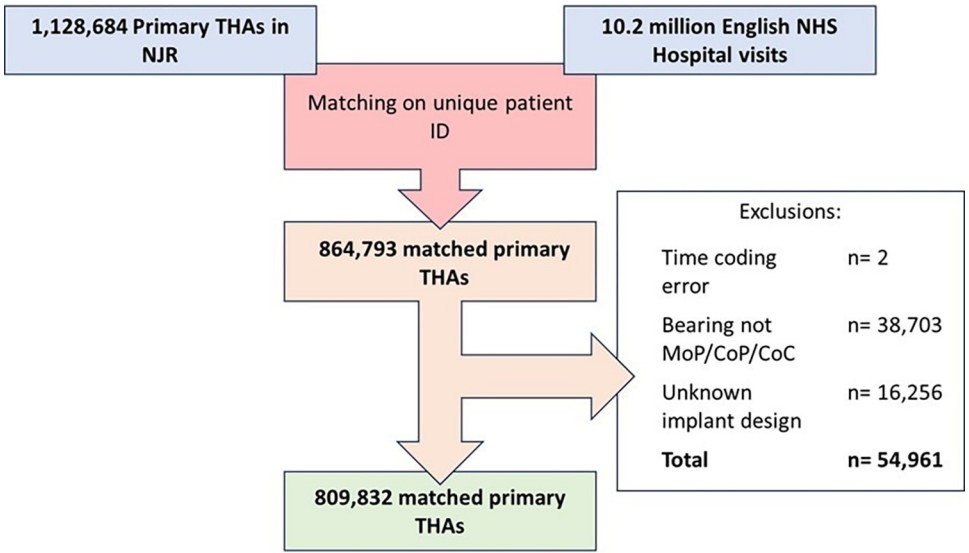

**Fig 1. Flow diagram demonstrating data sources, matching, and exclusions. NJR, National Joint Registry.**

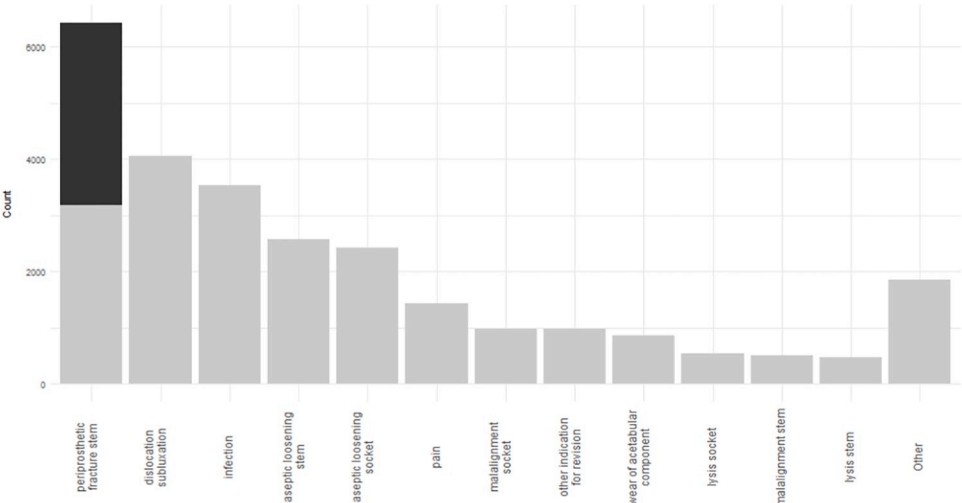

**Fig 2. Reasons for revision or reoperation of THRs.** Grey bars represent the revisions as documented by the NJR. The black area represents the additional POPFFs treated with internal fixation (black area) identified in this study. A breakdown of the indications for revision grouped within the "Other" category is provided in the S2_tab. NJR, National Joint Registry; POPFF, postoperative periprosthetic femoral fracture; THR, total hip replacement.

with the majority of patients being female, ASA 2, and undergoing THR for predominantly osteoarthritis [5]. The full table of demographics can be seen in Table 1.

Table 2 demonstrates the demographics (at the time of primary THR) that went on to sustain POPFF and underwent surgery for the fracture. These patients appeared to be older, with a higher ASA and had their primary surgery for indications other than solely osteoarthritis.

## Incidence

We observed that 22,647 of the 809,832 patients (2.8%) underwent surgery for revision (for any cause) or POPFF fixation, during a total of 5,542,332 patients years giving an incidence rate of 4.09 (95% CI 4.03, 4.14) reoperations (for revision or fixation) per 1,000 prostheses years, and 5,100 of the 809,832 included THRs (0.6%) underwent surgery for POPFF giving an incidence rate of 0.92 (95% CI 0.90, 0.95) POPFF per 1,000 prostheses years. The number of new cases of POPFF among patients who had a primary hip replacement recorded in the NJR increased each year (Fig 3).

## Analyses restricted by age and indication

There were 407,894 THR performed in patients over the age of 70, leading to 10,691 revision or fixation for POPFF over an exposure time of 2,613,659 years, giving an overall PTIR for revision for any cause or POPFF fixation of 4.09 (95% CI 4.01, 4.17) per 1,000 prostheses years, and 3,417 hips had surgically managed POPFF giving a PTIR for POPFF was 1.31 (95% CI 1.26, 1.35) per 1,000 prostheses years.

There were 94,543 primary THR performed for an indication other than osteoarthritis alone, leading to 3,582 revision or fixation for POPFF over an exposure time of 564,820 years, giving an overall PTIR for revision for any cause or POPFF fixation of 6.34 (95% CI 6.14, 6.55) per 1,000 prostheses years. A total of 786 patients had surgically managed POPFF giving a PTIR for POPFF was 1.39 (95% 1.30, 1.49) per 1,000 prostheses years.

There were 34,856 THR performed for neck of femur fracture, leading to 1,282 revision or fixation for POPFF over an exposure time of 173,650 years, giving an overall PTIR for revision

**Table 1. Demographics of overall cohort at time of primary joint replacement.**

| | Overall |
|---|---|
| **n** | 809,832 |
| **Age (years) (median [IQR])** | 71.00 [63.00, 77.00] |
| **Patient gender (%)** | |
| Female | 496,576 (61.3) |
| Male | 313,249 (38.7) |
| Non-binary | 7 (0.0) |
| **ASA at primary THA (%)** | |
| P1—Fit and healthy | 101,097 (12.5) |
| P2—Mild disease not incapacitating | 556,756 (68.7) |
| P3—Incapacitating systemic disease | 146,984 (18.1) |
| P4—Life threatening disease | 4,922 (0.6) |
| P5—Expected to die within 24 h with or without an operation | 73 (0.0) |
| **Ethnicity (%)** | |
| White | 688,336 (85.0) |
| Unknown | 109,167 (13.5) |
| Non white | 12,329 (1.5) |
| **Indication for primary THA (%)** | |
| Osteoarthritis | 715,289 (88.3) |
| Acute trauma including NOF | 34,856 (4.3) |
| AVN | 19,802 (2.4) |
| Chronic trauma | 8,802 (1.1) |
| Inflammatory arthritis | 10,918 (1.3) |
| Malignancy | 1,072 (0.1) |
| Other | 4,874 (0.6) |
| Paediatric disease | 13,347 (1.6) |
| Previous arthrodesis | 236 (0.0) |
| Previous infection | 636 (0.1) |
| **Operation funding (%)** | |
| Independent | 14,391 (1.8) |
| NHS | 783,225 (96.7) |
| Unknown | 12,216 (1.5) |

for any cause or POPFF fixation of 7.38 (95% CI 6.98, 7.80) per 1,000 prostheses years. A total of 380 patients had surgically managed POPFF giving a PTIR for POPFF of 2.19 (95% CI 1.97, 2.42) per 1,000 prostheses years.

## Timing of fracture

Fig 4 demonstrates that after an early peak, the rate of fracture is relatively consistent. At 10 years, the cumulative probability of sustaining a POPFF requiring surgery was 1% (95% CI 1.0, 1.0).

## Mortality

Of the 5,100 patients who had a POPFF identified, 2,347 died during the follow-up period of this study. Death within 30 days of operation to treat the fracture was 4.2% (95% CI 3.6, 4.7), 90 days was 6.7% (95% CI 6.0, 7.4), and within 1 year of injury was 15.4% (95% CI 14.4, 16.4) versus mortality following revision for non-periprosthetic fracture causes which was 0.9% (95% CI 0.8, 1.1) within 30 days, 1.2% (95% CI 1.0, 1.3) within 90 days, and 4.7% (95% CI 4.3,

**Table 2. Demographics of patients (at time of primary hip replacement) who went on to sustain a POPFF.**

| | No fracture | | POPFF | |
|---|---|---|---|---|
| n | 804,732 | | 5,100 | |
| **Age (mean (SD))** | 69.37 | (11.04) | 73.73 | (9.95) |
| **ASA (%)** | | | | |
| P1—Fit and healthy | 100,639 | (12.5) | 458 | (9.0) |
| P2—Mild disease not incapacitating | 553,397 | (68.8) | 3,359 | (65.9) |
| P3—Incapacitating systemic disease | 145,751 | (18.1) | 1,233 | (24.2) |
| P4—Life threatening disease | 4,872 | (0.6) | 50 | (1.0) |
| P5—Expected to die within 24 h with or without an operation | 73 | (0.0) | 0 | (0.0) |
| **Indication for primary THR (%)** | | | | |
| Acute trauma including NOF | 34,476 | (4.3) | 380 | (7.5) |
| AVN | 19,659 | (2.4) | 143 | (2.8) |
| Chronic trauma | 8,673 | (1.1) | 129 | (2.5) |
| Inflammatory arthritis | 10,861 | (1.3) | 57 | (1.1) |
| Malignancy | 1,066 | (0.1) | 6 | (0.1) |
| Osteoarthritis | 710,975 | (88.3) | 4,314 | (84.6) |
| Other | 5,705 | (0.7) | 41 | (0.8) |
| Paediatric disease | 13,317 | (1.7) | 30 | (0.6) |

POPFF, postoperative periprosthetic femoral fracture; THR, total hip replacement.

5.0) within 1 year. The median survival time following fracture was 6.2 years (95% CI 5.9, 6.6 years). A KM plot with death after fracture as the outcome can be seen in Fig 5.

## Length of stay

The median length of overall stay in the acute hospital was 16 days (IQR 10, 26) with patients waiting a median of 3 days prior to receiving surgery (IQR 1, 5). The distribution of length of stay can be seen in S1 Fig.

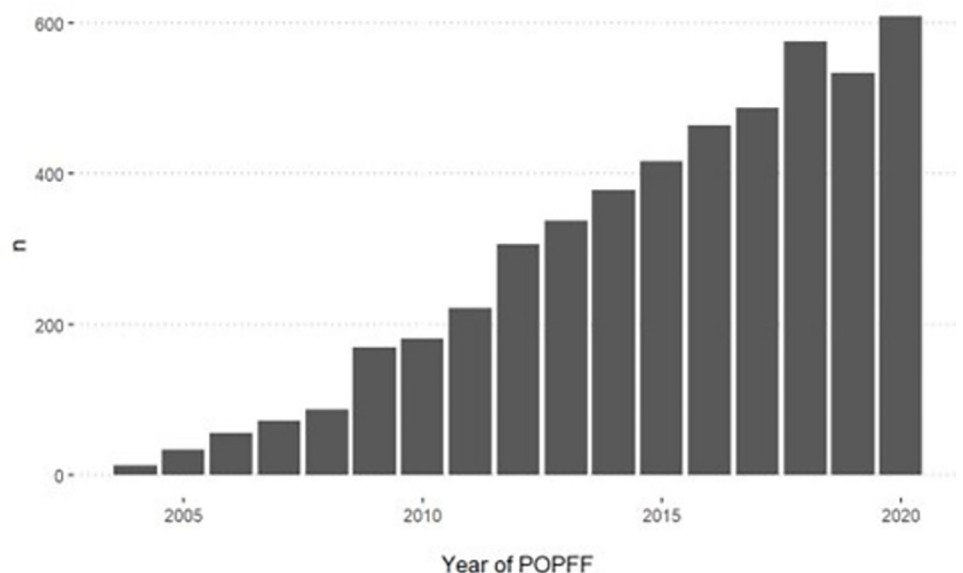

**Fig 3. Count of postoperative POPFF identified by year.** POPFF, postoperative periprosthetic femoral fracture.

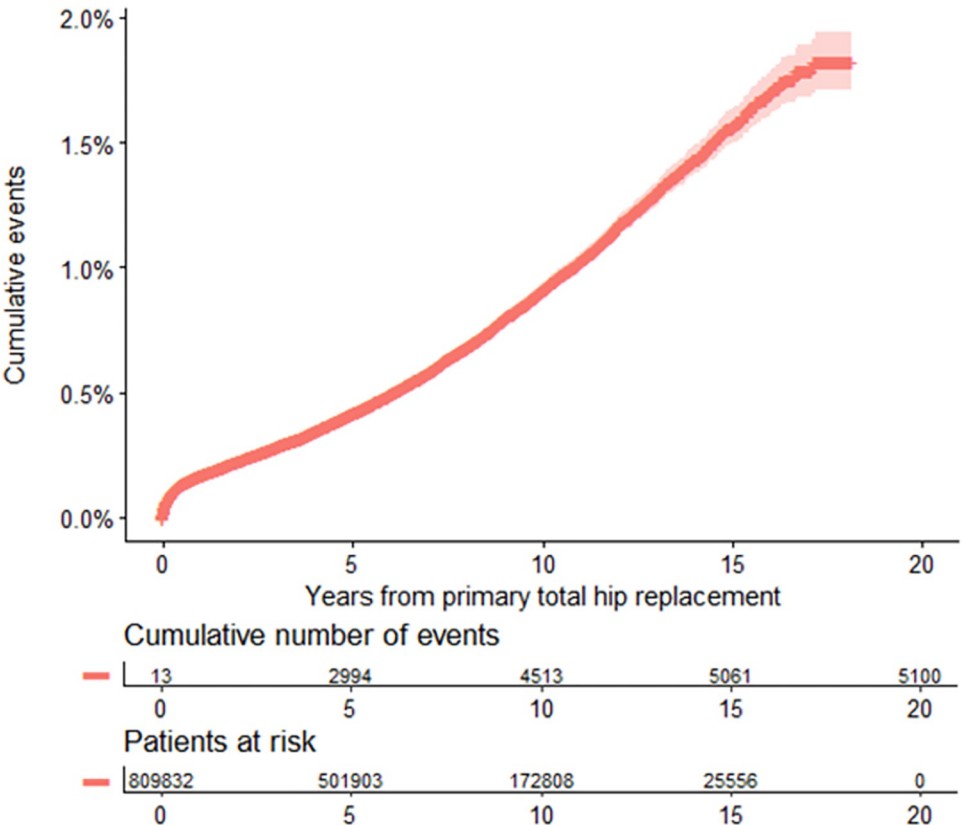

**Fig 4. Cumulative incidence plot showing occurrence of POPFF.** The shaded area indicates 95% confidence intervals of the unadjusted cumulative incidence estimate. POPFF, postoperative periprosthetic femoral fracture.

## Cost

Every year since 2015 has seen a total of over 10,000 acute hospital bed days per calendar year for the treatment of POPFF. The estimated cost of inpatient stay alone was £5,865,900 a year, not accounting for the surgical management of the injury, facilities, subsequent rehabilitation, and complications [16].

## Discussion

### Statement of principal findings

We have observed that in our study population, postoperative periprosthetic femoral fracture is the single most common indication for major reoperation following a total hip replacement. Previous estimates were based on revision surgery alone. We have demonstrated that 56% of fractures were not identified by the NJR, which does not capture those that are treated with fixation without revision of components.

Patients with POPFF were older and frailer than patients without POPFF and 90-day mortality for patients who have undergone revision or fixation of a POPFF was greater than 5 times more than for any other type of revision operation for hip replacement [17]. The cost of looking after patients with these injuries (in bed days alone) may be over £5 million a year without accounting for theatre time, equipment, follow up, and subsequent complications. These results emphasise the significant vulnerability of this patient group in relation to other groups of patients with problematic hip replacement.

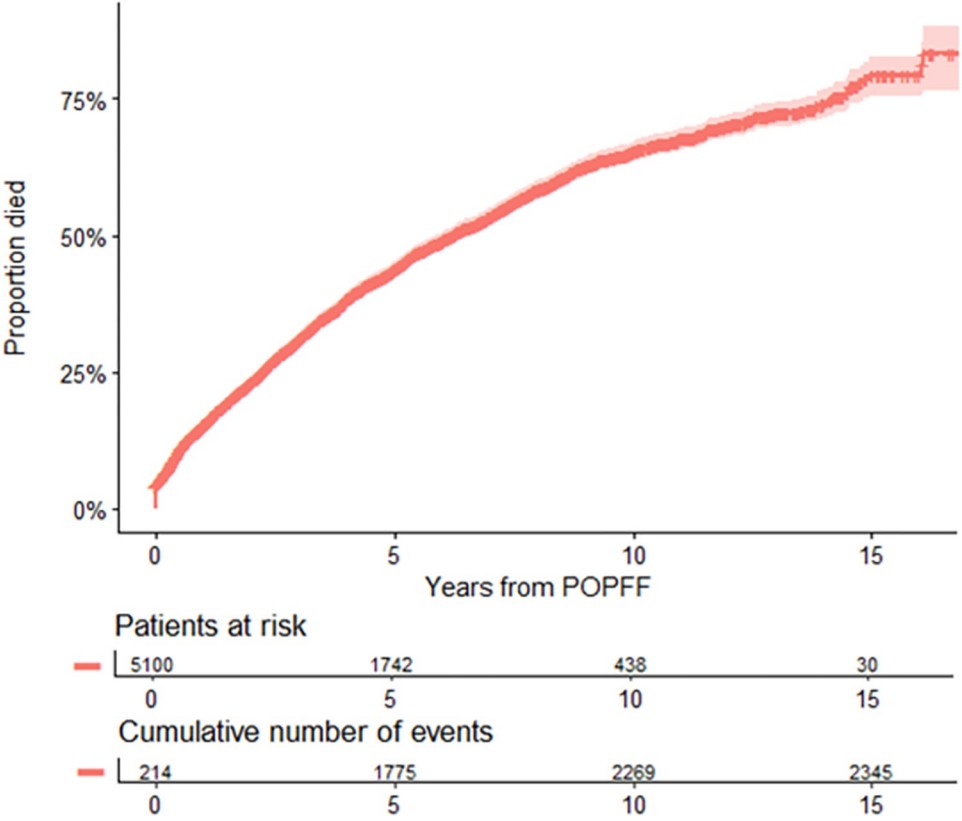

**Fig 5. Cumulative mortality (unadjusted) following POPFF.** Shaded area indicates the cumulative incidence estimate 95% confidence interval. POPFF, postoperative periprosthetic femoral fracture.

### Strengths and weaknesses of the study

This is the largest study of its kind identifying fractures around THR stems. The strength of this study is the size and documented completeness of the large mandatory national registry. Although registry data is now considered highly reliable, the initial years (2004 to 2008) of the NJR were less complete and there will be both primary and revision cases missing from these analyses. In this study, information from hospital coding data has been used to identify a greater number of revision operations, which may have previously been missed. These methods are likely to improve accuracy of the data and subsequent observations. We have successfully identified POPFF treated with internal fixation and added them to the already known revisions within the NJR but have not addressed the same issue for two other key reasons for failure infection and dislocation. Dislocated THRs treated with closed or open reduction without modification or changing of the implants will not be identified in this cohort, this is hard to assess using admitted patient HES data (as in this study) as many dislocations are treated with closed reduction in the Emergency Department and may not be admitted to hospital which will not be captured in the available linked data. Recent evidence has demonstrated that the cumulative incidence of dislocation may be as high as 0.9% within 30 days but as few as 11% of these patients undergo revision surgery for dislocation [18]. In a similar manner, infected THRs treated with a debridement and no exchange of implants were not captured prior to June 2018. Operations for infection including debridement and implant exchange will have been captured in the NJR data set. Reoperation for infection is associated with large

volume blood loss [19]. Addition of operations for dislocation and infection will improve the validity of large registry data sets in the future and allow for accurate planning and prioritisation of resource allocation and focus for future research. The codes used to identify POPFF in the linked database have not been formally validated as both sensitive and specific to identify only true cases as the pseudonymised data does not allow this. As a result, this may result in over or under identification of cases, the ratio of cases treated with revision and fixation is however consistent with data collected by the NHFD [7,9].

## Strengths and weaknesses of the study in relation to other studies

We estimate that the overall prevalence of periprosthetic fracture was 0.6% which was at least double that of previous estimates from large national registries [20–22]. These results are in keeping with previous smaller studies linking the Swedish Arthroplasty Register to national health data, which demonstrated that registry estimates alone under reported the incidence of surgically treated POPFF [11]. This study will not have identified patients who had POPFF treated without surgery, i.e., patients with a stable fracture pattern or those too unwell to tolerate surgery as we were reliant on OPCS4 operative codes to link the side of injury to an initial primary THR. These patients undoubtedly represent an important cohort and in the NHFD have been reported to represent approximately 20% of the total POPFF cohort. These extra patients may still require admission to hospital and will still require significant rehabilitation and social care support and need to be included in an overall health economic estimate of the burden of POPFF. This study identified an increase in the incidence of POPFF in the last decade; this must be interpreted with caution however as NJR data quality and compliance was less reliable in the first part of the last decade [11]. Although we have been unable to investigate the reasons behind delays in taking patients to theatre, this has in some way been addressed by the Facilities Audit of the 2021 National Hip Fracture Database, which cited surgeon availability as the largest cause for preoperative delay. An ongoing study by Imperial College London aims to investigate variations in care between hospitals and regions and further understand the metrics by which care of these patients should be judged [23]. This will be important in guiding further research in this area.

## Unanswered questions and future research

Previous failings in the identification of patients who have suffered periprosthetic fracture mean that this population has not been prioritised appropriately. Given the relative scale of the problem, focused efforts should be made to tackle POPFF with adaptation of surgical training, health care pathways, and funding for care and research. Future work using national data sets must also consider both unreported revision and fixation events to build a true picture of implant performance as well as POPFF treated without surgery. Implant survival estimates from the NJR may change once these additional reoperations are included as an endpoint of implant survival, and this effect should be explored in further work. A non-revision reoperation data collection form was added to the NJR in June 2023 but the uptake of this and capture rate is not yet known. As implant registers turn from implant monitoring tools to research tools, efforts should be made to increase the clinically important data captured so that results are reflective of the real-world patient experience.

Further work is needed to understand the causes of POPFF and whether it represents a true fragility fracture or is more related to implant-associated mechanical causes. Methods of preventing fractures such as bone protection and falls prevention may prove important and this warrants further investigation. We have focussed on fracture following THR, but this needs to be expanded to look at other replaced joints as well. The effect of POPFF on the patient

experience in the short and longer term requires further investigation, with reference to the specific outcomes of fixation and revision as treatment methods.

## Conclusion

Periprosthetic femoral fracture may be the largest cause for major re-operation following THR. Revision only data analysis appears to miss over half of patients who sustain this life changing injury if our method of data capture is accurate. POPFF is associated with high mortality and long stays in hospital both before and after surgery. Research is needed to improve care for this under researched patient group.

## Patient and public involvement

The pre-analysis plan for this project was approved by the patient representatives on the NJR research subcommittee which includes 2 patient representatives. The PEP-R group at the Musculoskeletal Research Unit at the University of Bristol advised on the primary outcome of interest and in the interpretation of the results, onward implications, and need for further research as well as aiding in the production of materials for dissemination of results.

## Supporting information

**S1 Table. Codes used in identifying fractures.**
(DOCX)

**S2 Table. Breakdown of reasons for revision including items contained within "Other".**
(DOCX)

**S1 Fig. Histogram depicting length of stay in the acute hospital after identification of post-operative periprosthetic femoral fracture (POPFF).**
(DOCX)

**S1 RECORD Checklist. Checklist indicating how this study met the RECORD reporting guidelines.**
(DOCX)

## Acknowledgments

We are grateful to the continued input and support of our patient group who have helped to shape the design and conduct of this study. We thank the patients and staff of all the hospitals who have contributed data to the National Joint Registry. We are grateful to the Healthcare Quality Improvement Partnership (HQIP), the NJR Research Committee and staff at the NJR for facilitating this work. The authors have conformed to the NJR's standard protocol for data access and publication.

The views expressed represent those of the authors and do not necessarily reflect those of the National Joint Registry Steering Committee, Research Subcommittee, or the Healthcare Quality Improvement Partnership (HQIP) who do not vouch for how the information is presented.

The views expressed in this publication are those of the author(s) and not necessarily those of the NHS, the National Institute for Health Research, or the Department of Health and Social Care.

The views expressed are those of the author(s) and not necessarily those of the NIHR or the Department of Health and Social Care.

## Author Contributions

**Conceptualization:** Jonathan Nicholas Lamb, Jonathan Thomas Evans, Michael Richard Whitehouse.

**Formal analysis:** Jonathan Nicholas Lamb, Jonathan Thomas Evans, Samuel Relton.

**Funding acquisition:** Jonathan Thomas Evans, Michael Richard Whitehouse.

**Investigation:** Jonathan Nicholas Lamb, Jonathan Thomas Evans, Samuel Relton, Michael Richard Whitehouse, J Mark Wilkinson, Hemant Pandit.

**Methodology:** Jonathan Nicholas Lamb, Jonathan Thomas Evans, Samuel Relton, Michael Richard Whitehouse, J Mark Wilkinson, Hemant Pandit.

**Project administration:** Jonathan Thomas Evans.

**Supervision:** Michael Richard Whitehouse, J Mark Wilkinson, Hemant Pandit.

**Visualization:** Jonathan Nicholas Lamb.

**Writing – original draft:** Jonathan Nicholas Lamb, Jonathan Thomas Evans.

**Writing – review & editing:** Jonathan Nicholas Lamb, Jonathan Thomas Evans, Samuel Relton, Michael Richard Whitehouse, J Mark Wilkinson, Hemant Pandit.

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
