## [Editor Report · Decision Letter 0]

11 Apr 2024

Dear Dr Evans, 

Thank you for submitting your manuscript entitled "Postoperative periprosthetic femoral fracture is the leading cause of reoperation following total hip replacement: An analysis of linked data from the National Joint Registry and Hospital Episodes Statistics." for consideration by PLOS Medicine.

Your manuscript has now been evaluated by the PLOS Medicine editorial staff and I am writing to let you know that we would like to send your submission out for external peer review.

Please re-submit your manuscript by the end of Monday 15th April. If you require more time, please just let me know.

Feel free to email our support team at plosmedicine@plos.org if you need help with inputting your metadata, etc. Otherwise, please feel free to email me directly on ssunny@plos.org

Kind regards,

Syba Sunny, MBBS, MRes, FRCPath

Associate Editor

PLOS Medicine

ssunny@plos.org

---

## [Decision Letter · Decision Letter 1]

16 May 2024

Dear Dr. Evans,

Thank you very much for submitting your manuscript "Postoperative periprosthetic femoral fracture is the leading cause of reoperation following total hip replacement: An analysis of linked data from the National Joint Registry and Hospital Episodes Statistics." (PMEDICINE-D-24-01147R1) for consideration at PLOS Medicine. 

The paper has been reviewed by two subject experts and a statistician; their comments are included below and can also be accessed here:

[LINK]

As you will see, the reviewers were positive about the paper but, they raised a number of questions about specific study details and the interpretation of data. After discussing the paper with the editorial team, I’m pleased to invite you to revise the paper in response to the reviewers’ comments. We plan to send the revised paper to some or all of the original reviewers*, and of course we cannot provide any guarantees at this stage regarding publication.

When you upload your revision, please include a point-by-point response that addresses all of the reviewer and editorial points, indicating the changes made in the manuscript and either an excerpt of the revised text or the location (e.g. page and line number) where each change can be found. Please submit a clean version of the paper as the main article file and a version with changes marked should as a marked-up manuscript. Please also check the guidelines for revised papers at http://journals.plos.org/plosmedicine/s/revising-your-manuscript for any that apply to your paper.

We ask that you submit your revision by Jun 06 2024 11:59PM. However, if this deadline is not feasible, please contact me by email, and we can discuss a suitable alternative.

Please don’t hesitate to contact me directly with any questions (ssunny@plos.org). If you reply directly to this message, please be sure to ‘Reply All’ so your message comes directly to my inbox.

Kind regards,

Syba

Syba Sunny MBBS, MRes, FRCPath

Associate Editor 

PLOS Medicine

ssunny@plos.org

*Please note: If your article is accepted, you may have the opportunity to make the peer review history publicly available. The record will include editor decision letters (with reviews) and your responses to reviewer comments. If eligible, we will contact you to opt in or out.

Editorial comments:

1) The editorial team agree that the authors should be commended on undertaking research on an important complication of total hip replacements, and we are grateful for the opportunity to consider your work. However, we do agree with the reviewers’ points with regards to the handling and reporting of data, and also the need to further justify some of the assertions made. All reviewer comments will need to be addressed in full before we can consider the manuscript further.

2) Financial Disclosures

Please provide further information about the funding body, including the funder’s name in full. Also, please describe any affiliations or contracts with the National Joint Registry, if applicable.

3) Data Availability

Thank you for including a statement regarding data availability. Please also include a statement about accessing Hospital Episode Statistics data.

4) Reporting guidance

Please ensure that the study is reported according to the RECORD (STROBE if you feel it is more appropriate) guideline and include the completed RECORD checklist as Supporting Information. Please add the following statement, or similar, to the Methods: "This study is reported as per the Reporting of studies Conducted using Observational Routinely-collected health Data (RECORD) Statement (S1 Checklist)."

The relevant guidance can be found here: https://www.equator-network.org/reporting-guidelines/

5) Statistical reporting

Please quantify the main results with 95% CIs and p values.

When reporting p values please report as <0.001 and where higher as p=0.002, for example. When reporting 95% CIs please separate upper and lower bounds with commas instead of hyphens as the latter can be confused with reporting of negative values.

Please include the actual amounts and/or absolute risk(s) of relevant outcomes (including NNT or NNH where appropriate), not just relative risks or correlation coefficients. (example for absolute risks: PMID: 28399126).

Please include any important dependent variables that are adjusted for in the analyses.

6) Prespecified analysis plan/study protocol

Did your study have a prospective protocol or analysis plan? Please state this (either way) early in the Methods section.

For all observational studies, in the manuscript text, please indicate: (1) the specific hypotheses you intended to test, (2) the analytical methods by which you planned to test them, (3) the analyses you actually performed, and (4) when reported analyses differ from those that were planned, transparent explanations for differences that affect the reliability of the study's results. If a reported analysis was performed based on an interesting but unanticipated pattern in the data, please be clear that the analysis was data-driven.

7) Abstract layout

Please structure your abstract using the PLOS Medicine headings (Background, Methods and Findings, Conclusions), i.e. the Methods should be combined with Findings to make one section.

8) Author summary

At this stage, we ask that you include a short, non-technical Author Summary of your research to make findings accessible to a wide audience that includes both scientists and non-scientists. The authors summary should consist of 2-3 succinct bullet points under each of the following headings:

• Why Was This Study Done? Authors should reflect on what was known about the topic before the research was published and why the research was needed.

• What Did the Researchers Do and Find? Authors should briefly describe the study design that was used and the study’s major findings. Do include the headline numbers from the study, such as the sample size and key findings. 

• What Do These Findings Mean? Authors should reflect on the new knowledge generated by the research and the implications for practice, research, policy, or public health. Authors should also consider how the interpretation of the study’s findings may be affected by the study limitations. In the final bullet point of ‘What Do These Findings Mean?’, please describe the main limitations of the study in non-technical language.

The Author Summary should immediately follow the Abstract in your revised manuscript. This text is subject to editorial change and should be distinct from the scientific abstract. Please see our author guidelines for more information: https://journals.plos.org/plosmedicine/s/revising-your-manuscript#loc-author-summary

9) Introduction layout

Please address past research and explain the need for and potential importance of your study. Indicate whether your study is novel and how you determined that. If there has been a systematic review of the evidence related to your study (or you have conducted one), please refer to and reference that review and indicate whether it supports the need for your study.

10) Discussion layout

Please present and organize the Discussion as follows: a short, clear summary of the article's findings; what the study adds to existing research and where and why the results may differ from previous research; strengths and limitations of the study; implications and next steps for research, clinical practice, and/or public policy; one-paragraph conclusion.

Comments from the reviewers:

Reviewer #1: 

Realising that NJR data only captures revision procedures, this study linked data from the NJR to Hospital Episode Statistics to also detect re-operations for post-operative periprosthetic fracture that were treated by fracture fixation without revision. They found over half of re-operations were fracture fixations and therefore not included in the NJR. Further analysis of patient demographics, timing of fracture and cost of treatment is included.

The messages from this study are important and highlight a number of concerns about POPFF, however the paper has a number of areas in which it could be improved.

The major problem is that the authors state (in multiple places in the paper) that this study is of the "true" incidence of POPFF, but it is not, as they also point out that some POPFF are treated non-operatively and these are not included. This paper therefore studies only re-operation for POPFF, and this needs to be made clear in the manuscript.

A second major problem is that re-operation for POPFF is compared to only revision for other causes (such as dislocation and infection), so the claim that POPFF is the major cause of re-operation may be incorrect. While there is some discussion about this point, the comparison is unbalanced, and should be better qualified or removed. This consideration also needs to be applied to comparisons of demographic groups and mortality. 

Other criticisms include:

Introduction:

Common use descriptions (such as thigh bone, re-do surgery) should be removed as this is a scientific manuscript not a lay article.

The aims of the study need to be more clearly stated.

Methods: 

The inclusion/exclusion criteria need to be clearly stated in one place. Some of this information is in the first paragraph and some in the second.

How incidence of POPFF is calculated needs to be stated.

It is unclear what sensitivity analyses were carried out.

There is no mention of how hospital LOS data were obtained or information on hospital costs.

The patient/public involvement would be better placed toward the end in the acknowledgement section.

Results:

In this section the results should be clearly stated and interpretation left for the discussion. For example, comments about previous studies using NJR data, or bimodal distribution of fractures.

Figure 2 is the "reasons for revision" not "failure", as some revisions are carried out with a well-functioning prosthesis, such as for infection. Also what is the difference between the 7th reason for revision "other indication for revision" and the last reason "other"? This figure "shows" the frequency of each reason for revision, the reader "visualises". Part 2 of this should be removed for the reason above.

Similarly, the word "failure" should be removed from the total incidence rate per 1000 patient years. These are re-operations not failures. This is also a mis-leading calculation as it does not include all re-operations.

Figure 3 shows the number of POPFF by year, not the incidence as stated in the text.

The term PTIR needs definition.

I do not understand how the calculated overall incidence of revision and fracture fixation per 1000 patient years is the same as for those over 70 at time of THR (4.09 for both). I would expect these to be different. A comparison of those 70 and over with those less than 70 would be helpful.

Comparison by primary diagnosis may be clearer if grouped into OA, Fractured NOF and Other.

The timing of fracture treatment section needs to simply state the result.

Figure 4 should be titled the cumulative incidence of POPFF.

The display of mortality after fracture treatment may be more easily interpreted if shown as survival. Additionally, a comparison of survival after fracture treatment to survival after revision for other causes in figure 5 would be helpful.

The calculation of cost is difficult to follow without knowing the mean LOS. If in 2015 for instance the number of POPFFs was 410 (as shown graphically), and the median LOS is used, the result is far short of 10,000 hospital bed days. The LOS needs to be at the upper confidence level of 26 to achieve this.

Discussion:

The statement about POPFF being the greatest reason for re-operation needs to be qualified.

I don't understand how frequency, mortality and cost results make this group "vulnerable".

In the discussion there should be mention that only 80% of revisions for POPFF were captured in the NJR, and the implications of this.

The discussion should include the timing of the fracture and how it differs from previous presumptions.

The discussion could include mention of the patient impact.

I don't understand how patients with POPFF are "underserved". They all would have been adequately treated.

Implant survival estimates in the NJR will not change by adding fracture fixation data, as the implant will still be un-revised.

The discussion lacks detail of how this study compares to previous studies looking at POPFF with only one mentioned.

It may be helpful to the reader to have a comparison of the groups treated by revision +/- fixation to those treated by fixation alone. 

There is no discussion of prosthesis characteristics for those treated for a POPFF.

Reviewer #2: 

Thank you for opportunity to review this interesting paper. I think that the main findings highlight the previously under-estimated burden of periprosthetic fractures. However, in my opinion, the manuscript need some reconstruction. I would also prefer line numbers in the manuscript to make the communication easier.

Specific comments:

Abstract: Please report confidence intervals for incidences.

Introduction: 

In general I would prefer shorter introduction. Some parts may be considered to shortened or moved to the discussion. Some previous epidemiology of the periprosthetic femoral fractures should be included, including the estimated incidence and predisposing factors.

"In 2019, the National Hip Fracture Database (NHFD) started collecting information on POPFF but these data are crude, do not include implant information and do not yet appear to be universally completed.1" Please define "universally complete"? What is missing?

If you cite Bottle et al., please shortly repeat their main findings, not only some part of discussion and weaknesses of their study.

Methods

Please define all outcomes you are reporting in the results. Now the results section is very heterogeneous and does not match the outcomes defined. For example, some calculation of the costs are reported in the results but not once mentioned in the methods.

Statistical analysis-paragraph includes mostly information that should not be under statistics-section in my opinion. Please restructure the methods section and leave only part that consider the statistical analyses here.

Results

Figure 2 highlights the missing POPFFs in the NJR data. However, in the limitations of the study you mention that THA infections treated with debridement without change of components are not recorded in the data. Therefore, if I have understood correctly, this figure gives false comparison of the burden of POPFFs compared with infections. Also, the dislocations treated with closed reposition are missing. I find this figure misleading and would consider removing or altering it. If you decide to keep this figure, this problem should be clearly mentioned in the discussion.

"The incidence of POPFF amongst patients who had a primary hip replacement recorded in the NJR appears to be rising by year (Figure 3). This figure must be interpreted with caution however as the early part of the chart represents a time where not as many hips had been input into the NJR and were therefore at risk of POPFF"

This type of speculation inn valid, however it should be in discussion, not in results.

"It is commonly considered that POPFF occur early after implantation (and may represent a missed intraoperative fracture) or that they occur late after implantation (e.g. 10 years or more) secondary to loosening and/or lysis.15" Never use references in the results section, again, this to the introduction or to discussion.

Demographics: Start the result section with demographics and give it short.

Mortality: The mortality rate is known to be high in periprosthetic fractures. Could you compare the mortality rates between patient who suffered a POPFF and those who did not?

Discussion

The first paragraph should focus on the main finding of this study. The following part of the first paragraph could be moved to later discussion and partly added in the conclusions: "A non-revision reoperation data collection form was added to the NJR in June 2023 but the uptake of this and capture rate is not yet known. 90-day mortality for patients who have undergone revision or fixation of a POPFF was greater than five times more than for any other type of revision operation for failing hip replacement.5 The cost of looking after patients with these injuries (in bed days alone) may be over £5 million a year without accounting for theatre time, equipment and follow up/subsequent complications. These results emphasise the significant vulnerability of this patient group in relation to other groups of patients with problematic hip replacement."

Please compare your results in previous studies. What has been the estimated incidence of POPFFs inn the previous studies?

Conclusions: The economic burden could be mentioned here, but not in the principal findings of the study since this was not a cost-analysis study.

Reviewer #3: 

Using data from the English(?) National Joint Registry and the National Health Service, the incidence of post-operative periprosthetic femoral fracture after total hip replacement from 2004-2020 inclusive was estimated. 

1. Please state what country's national joint registry is being used in the title and in the Background section of the Abstract.

2. Were the 5100 POPFF identified as having been treated with revision surgery or internal fixation those identified in the NJR only? Were the 2831 fractures those identified in the NHS data only? Were any fractures identified in both datasets? That is, please clarify that the 2831 fractures are not included in the NJR dataset. 

3. Figure 2 - please state which figure corresponds to before and after inclusion of POPFF in the caption. Is the only difference between the top and bottom histogram the addition of the 2831 fractures identified in the NHS data? 

4. Please include the number of patient-years the 809,832 included THRs represents when describing incidence. 

5. What is PTIR? This acronym appears first just after Figure 3 but is not defined. 

6. What is the shaded area on Figure 4? Please include counts of THAs and events at 0, 5, 10, 15 and 20 years below this figure. 

7. The patient demographics section just after Figure 4 should appear as the first subsection of the Results. The Table of demographics of the overall cohort should appear in the main paper. 

8. Table 1 provides p-values assessing the differences in various demographic factors for patients who developed POPFF or not. I don't think that Table 1 is particularly useful - if interest is in knowing what demographic factors are causally related to the development of a POPFF, then confounding of the relationship between each of these factors and the outcome must be considered in turn, adjusting for other factors to reduce confounding of this relationship (and a causal diagram should be constructed to aid in selection of these factors). I recommend deleting this and all discussion of this Table. (For an in-depth discussion of this issue, I recommend Hernan and Robins' textbook What if?) Also note that when p-values are presented, it is essential to state which statistical tests were used to generate these. 

9. In Figure 5, include the sample size at each year along with cumulative number of events. Please also state what the pink shading is in the caption. 

10. How many hospital stays in the Length of Stay section were concluded with discharge from hospital, and how many with in-hospital mortality? 

Minor comments:

1. Introduction: please change "patient's report a loss" to "patients report a loss"

2. When it is stated in the Introduction that the NJR is linked to itself, does this mean that records from the same patient are linked to each other?

3. Please provide a reference for the statement "There are 520 additional revisions… this is roughly in keeping with results of the NJR annual data quality audits."

[LINK]

1. Please upload any figures associated with your paper as individual TIF or EPS files with 300dpi resolution at resubmission; please read our figure guidelines for more information on our requirements: http://journals.plos.org/plosmedicine/s/figures. While revising your submission, please upload your figure files to the PACE digital diagnostic tool, https://pacev2.apexcovantage.com/. PACE helps ensure that figures meet PLOS requirements. To use PACE, you must first register as a user. Then, login and navigate to the UPLOAD tab, where you will find detailed instructions on how to use the tool. If you encounter any issues or have any questions when using PACE, please email us at PLOSMedicine@plos.org.

To submit your revised manuscript please use the following link:

---

## [Decision Letter · Decision Letter 2]

3 Jul 2024

Dear Dr Evans,

Many thanks for submitting your revised manuscript, entitled "Postoperative periprosthetic femoral fracture is the leading cause of major reoperation following total hip replacement: An analysis of linked data from the United Kingdom based National Joint Registry and Hospital Episodes Statistics." (PMEDICINE-D-24-01147R2) to PLOS Medicine. The paper has been re-reviewed by a subject expert and a statistician; their comments are included below and can also be accessed here: [LINK]

As you will see, the reviewers were, on the whole, satisfied with your revised manuscript. However, an academic editor with relevant expertise was also consulted and a concern was raised (see below). Your revised paper was also discussed with the wider editorial team; all were in agreement with the academic editor. As such, we invite you to submit a further revised manuscript addressing this issue and further comments below. Please note that we cannot provide any guarantees at this stage regarding publication.

We ask that you submit your revision by Jul 29 2024 11:59PM. However, if this deadline is not feasible, please contact me by email, and we can discuss a suitable alternative.

Don't hesitate to contact me directly with any questions (ssunny@plos.org). 

Best regards, 

Syba 

Dr Syba Sunny, MBBS, MRes, FRCPath 

Associate Editor

PLOS Medicine

ssunny@plos.org

Editorial requests:

We thank you for your thorough revision and we continue to find your research interesting and of clinical importance. As mentioned previously, we invite you to submit a revised manuscript. 

We ask that you address the academic editor's comments (below) in full; we believe she has highlighted a significant issue. It would be beneficial to state whether the strategy used to identify patients with postoperative periprosthetic femoral fracture (POPFF) was validated and, if not, we ask that you make an acknowledgement of this in your main text. 

Considering the limitations of the data, we also ask if you would consider re-wording the conclusions asserted in all relevant sections to take into account that the incidence of POPFF as a complication of total hip replacement may not be as significant as stated in this and the original version of the manuscript.

Finally, there has been some comments about Figure 2 and the lack of inclusion of counts for 'other' causes of revision that might have been missed in the NJR for what appears to be similar reasons as with POPFF. We think that the readers would appreciate reading the counts for this and this might help them to better contextualise POPFF amongst other complications of total hip replacements. We ask that you address this point further in your second revision.

Thanks again.

Comments from the academic editor:

The academic editor believed that the topic was 'interesting and clinically relevant' (in keeping with the sentiments of the reviewers and editors). However, she was concerned that the incidence of POPFF might be overestimated by the methods you have used here. She asked that the reliability of the combination of codes used to identify POPFF be clarified and the risk of misdiagnosis (with differential diagnoses) using these codes be discussed as a limitation. 

Comments from the reviewers: 

Reviewer #1: A much improved manuscript with greater clarity. 

Some simple grammar corrections required. 

Perhaps figure 2 could be simplified by discarding the part 2 and a hatched portion added to the periprosthetic fracture column rather than repeating the whole histogram with only change.

Reviewer #3: I thank the authors for their responses to my comments on the previous version of this manuscript. I have no further comments.

---

## [Editor Report · Decision Letter 3]

14 Aug 2024

Dear Dr. Evans,

Thank you very much for re-submitting your manuscript "Postoperative periprosthetic femoral fracture is the leading cause of major reoperation following total hip replacement: An analysis of linked data from the United Kingdom based National Joint Registry and Hospital Episodes Statistics." (PMEDICINE-D-24-01147R3) for review by PLOS Medicine.

I have discussed the paper with my colleagues and the academic editor; I am pleased to say that, provided the remaining editorial and production issues are dealt with, we are now planning to accept the paper for publication in the journal.

The remaining issues that need to be addressed are listed at the end of this email. 

In your rebuttal letter, you should indicate your response to the editors' comments and the changes you have made in the manuscript. Please submit a clean version of the paper as the main article file. A version with changes marked must also be uploaded as a marked up manuscript file.

Please also check the guidelines for revised papers at http://journals.plos.org/plosmedicine/s/revising-your-manuscript for any that apply to your paper. 

We look forward to receiving the revised manuscript by Aug 21 2024 11:59PM. Please do let us know if you need more time. 

Sincerely,

Syba

Syba Sunny, MBBS, MRes, FRCPath

Associate Editor 

PLOS Medicine

ssunny@plos.org

Requests from Editors:

Thank you for engaging so thoroughly with previous editor and reviewer comments. We have further comments/requests; however, most of these simply pertain to journal-specific formatting and content requirements. We ask that you address all in full in order to progress towards publication.

COMPETING INTERESTS

Thank you for the information provided in the Competing Interests section. We very much appreciate as much transparency as possible at PLOS, so we appreciate the authors’ efforts here. I only have one small ask – could you replace the initials ZB with the name of the company these represent, please?

TITLE

Please revise your title according to PLOS Medicine's style. Your title must be nondeclarative. Causality can be inferred only for an RCT. It could perhaps be changed to something like ‘The incidence of periprosthetic femoral fracture following total hip replacement: an analysis of UK National Joint Registry and Hospital Episodes Statistics data’, or similar, as you see fit. 

ABSTRACT

Thank you for amending your abstract as requested previously. I note that this amended version hasn’t come across to the relevant ‘meta data’ section (though it is present in your main text) – could you have a look at this and correct this please? If this is not something you find you can correct, please link in with our Editorial Office who might be able to help; they can be contacted via plosmedicine@plos.org.

Also, please remove your funding statement from this section – this can be removed and transferred to the Financial Disclosure section (which forms part of the meta-data for the manuscript).

AUTHOR SUMMARY

Thank you for all your work on this section. Could you kindly address 3 small points in this section:

(1) In the section ‘What did the researchers do and find?’, please revise the sentence ‘When cases treated with revision and fixation are combined, periprosthetic fracture is the most common…’ so it reads ‘… periprosthetic fracture appears to be the most common…’

(2) Could you rename the section ‘ to ‘What Do These Findings Mean?’

(3) In this last section, please revise the sentence ‘These results demonstrate that periprosthetic fracture after hip replacement is the biggest problem facing patients with a hip replacement…’ to ‘These results suggest that …’ or similar.

TABLES and FIGURES

Throughout, including the supporting files, please provide titles/captions/footnotes which clearly describe the table/figure content without the need to refer to the text.

Please ensure all abbreviations including those used for statistical reporting are also clearly defined in the footnote.

Throughout please indicate whether your analyses are adjusted or unadjusted and where adjusted analyses are presented please also present unadjusted analyses for comparison. 

Please also ensure to clearly detail in the footnote/caption the factors which are adjusted for.

Please refer to https://journals.plos.org/plosmedicine/s/figures#loc-pages for further guidance.

SOCIAL MEDIA

To help us extend the reach of your research, if not already done so, please detail any X (formerly Twitter) handles you wish to be included when we tweet this paper (including your own, your coauthors’, your institution, funder, or lab) in the manuscript submission form when you re-submit the manuscript.

---

## [Editor Report · Decision Letter 4]

16 Aug 2024

Dear Dr Evans, 

On behalf of my colleagues and the Academic Editor, I am pleased to inform you that we have agreed to publish your manuscript "The incidence of periprosthetic femoral fracture following total hip replacement: an analysis of UK National Joint Registry and Hospital Episodes Statistics data." (PMEDICINE-D-24-01147R4) in PLOS Medicine.

PRESS

Sincerely, 

Syba

Syba Sunny, MBBS, MRes, FRCPath 

Associate Editor 

PLOS Medicine